# Prognostic Value of HER2-Low Expression in Non-Metastatic Triple-Negative Breast Cancer and Correlation with Other Biomarkers

**DOI:** 10.3390/cancers13236059

**Published:** 2021-12-01

**Authors:** William Jacot, Aurélie Maran-Gonzalez, Océane Massol, Charlotte Sorbs, Caroline Mollevi, Séverine Guiu, Florence Boissière-Michot, Jeanne Ramos

**Affiliations:** 1Department of Medical Oncology, Montpellier Cancer Institute Val d’Aurelle, 208 rue des Apothicaires, F-34298 Montpellier, France; Charlotte.Sorbs@icm.unicancer.fr (C.S.); Severine.Guiu@icm.unicancer.fr (S.G.); 2Translational Research Unit, Montpellier Cancer Institute Val d’Aurelle, 208 rue des Apothicaires, F-34298 Montpellier, France; florence.boissiere@icm.unicancer.fr (F.B.-M.); Jeanne.Ramos@icm.unicancer.fr (J.R.); 3Faculty of Medicine, Montpellier University, F-34090 Montpellier, France; 4Institut de Recherche en Cancérologie de Montpellier (IRCM), Inserm U1194, F-34298 Montpellier, France; aurelie.maran-gonzalez@icm.unicancer.fr; 5Department of Pathology, Montpellier Cancer Institute Val d’Aurelle, 208 rue des Apothicaires, F-34298 Montpellier, France; 6Biometrics Unit, Institut du Cancer Montpellier (ICM), Université de Montpellier, 208 rue des Apothicaires, F-34298 Montpellier, France; oceane.m@dbmail.com (O.M.); Caroline.Mollevi@icm.unicancer.fr (C.M.)

**Keywords:** triple-negative breast cancer, HER2-low, expression, prognosis, molecular apocrine, androgen receptor

## Abstract

**Simple Summary:**

HER2-low breast cancer is an emerging subtype for which very few data are available, especially within the triple-negative breast cancer (TNBC) group. Our aim was to evaluate HER2 expression using the 2018 ASCO/CAP guidelines for HER2 scoring and its prognostic value in a large retrospective series of 296 patients with non-metastatic TNBC. In this large, homogeneous series of patients with early TNBC, HER2 1+/2+ expression was associated with a lower histological grade and molecular apocrine profile, and a specific subgroup of androgen receptor-expressing TNBC with worse long-term prognosis. Considering the current development of anti-androgens for molecular apocrine-like tumors and anti-HER2 antibody–drug conjugates for this HER2 1+/2+ population, this clinicopathological association could have therapeutic implications. Although rare (2.7% of patients in our series), HER2 2+ TNBC seems to have worse relapse-free survival, advocating a dedicated clinical, biological and therapeutic evaluation of this subgroup.

**Abstract:**

HER2-low breast cancer (i.e., HER 1+ or 2+, without gene amplification) is an emerging subtype for which very few data are available, especially within the triple-negative breast cancer (TNBC) group. Our aim was to evaluate HER2 expression and its prognostic value in a large retrospective series of patients with non-metastatic TNBC (median age: 57.7 years; range: 28.5–98.6). Among the 296 TNBC samples, 83.8% were HER2 0, 13.5% were HER2 1+, and 2.7% were HER2 2+ (HercepTest^TM^ and 2018 ASCO/CAP guidelines for HER2 scoring). CK5/6 and/or EGFR-expressing androgen receptors and FOXA1-expressing tumors were classified as basal-like (63.8%) and molecular apocrine-like (MA, 40.2%), respectively. Compared with HER2 0 tumors, HER2 1+/2+ tumors exhibited a lower histological grade (1/2) (35.4% vs. 18.2%, *p* = 0.007) and MA profile (57.5% vs. 36.7%, *p* = 0.008). Moreover, patients with HER2 1+/2+ tumors were older (*p* = 0.047). After a median follow-up of 9.7 years, HER2 2+ tumors (compared with HER2 0/1+ tumors) were associated with worse relapse-free survival (RFS) (HR = 3.16, 95% CI [1.27; 7.85], *p* = 0.034) in a univariate analysis. Overall survival (OS) and RFS were not different in the HER2 0 and 1+/2+ groups. HER2 levels were not significantly associated with OS or RFS in a multivariate analysis.

## 1. Introduction

Triple-negative breast cancer (TNBC) represents 15% of all breast cancer types, and is defined by the lack of significant estrogen receptor (ER) and progesterone receptor (PR) expression, and the absence of human epidermal receptor 2 (HER2) overexpression/amplification [1,2]. Despite its chemosensitivity, TNBC is associated with poor prognosis [1,3].

Currently, HER2 status is routinely assessed in all patients with breast cancer, by immunohistochemistry (IHC) and/or in situ hybridization (ISH). According to the American Society of Clinical Oncology (ASCO)/College of American Pathologists (CAP) 2018 recommendations [4], tumors are considered to be HER2-positive (HER2+) when HER2 overexpression is observed by IHC (score 3+, strong expression), or when *ERBB2* gene amplification is detected by fluorescence in situ hybridization (FISH). Tumors with a score of 0 and 1+ (weak expression) are considered to be HER2-negative, and tumors with a score of 2+ (moderate expression) need to be assessed by FISH for confirmation. HER2 1+ and HER2 2+ tumors with negative FISH are now classified as HER2-low tumors (~45–55% of breast cancers) [5,6,7]. In TNBC, HER2 levels can vary from absent (score of 0) to moderate expression without *ERBB2* gene amplification (score of 2+ ISH−).

The efficacy of anti-HER2 agents in HER2-low tumors has not been demonstrated yet [8,9], and this breast cancer subgroup is thought to have a poorer prognosis compared with HER2 0 tumors, based on retrospective studies. However, the published evidence shows conflicting results from one clinical setting to another one, and depending on hormone receptor status [8,10,11,12].

New anti-HER2 antibody–drug conjugates (ADC), which combine an anti-HER2 antibody and a cytotoxic agent, seem to be effective in some HER2-low tumors; for instance, trastuzumab deruxtecan, a new anti-HER2 ADC that combines trastuzumab and a topoisomerase 1 inhibitor, showed promising results in heavily pretreated patients with metastatic HER2-low breast cancer (HER 1+/2+) [13]. On the basis of these first results, an ongoing phase III study was started to compare this ADC with conventional chemotherapy (ClinicalTrials.gov, accessed on 15 October 2021, Identifier: NCT03734029). If these preliminary results are confirmed, this new approach could be added to the therapeutic arsenal for this subgroup of patients with TNBC, to whom only chemotherapy can currently be proposed. However, the biology of breast cancers expressing low HER2 levels remains poorly investigated, especially in the TNBC group.

In this study, we evaluated the impact of HER2 expression level (0, 1+, 2+) and its association with clinicopathological features and survival outcomes in a cohort of 296 patients with early TNBC treated at our institution, previously characterized for emerging biomarkers [14,15,16,17].

## 2. Results

### 2.1. Patient and Tumor Characteristics

For the present study, clinicopathological data on 296 consecutive patients with TNBC, referred to our institution between 2002 and 2012, were extracted from a previously published patient database [14,15,16,17]. Table 1 lists their main clinicopathological characteristics that were consistent with classical TNBC features. The patients’ median age was 57.7 years (range: 28.5–98.6 years). Ductal carcinoma was the most common histological type (81.9%), 75.3% of the patients received adjuvant chemotherapy, and the remaining 24.7% received adjuvant radiation therapy if clinically indicated. None of the included patients received hormonal therapy, targeted therapy, or an investigational product.

### 2.2. HER2 Expression and Pathological Associations

Among the 296 TNBC samples, 248 (83.8%) were classified as HER2 0, 40 (13.5%) as HER2 1+, and 8 (2.7%) as HER2 2+. On the basis of the HER2 expression level distribution, correlation analyses were performed by classifying tumors into two groups (HER2 0 vs. HER2 1+/2+).

HER2 1+/2+ tumors were significantly more frequent in older patients, and displayed a lower histological grade and a molecular apocrine phenotype more frequently compared with HER2 0 tumors (Table 2). No statistical correlation was found between the other emergent biomarkers previously evaluated in this cohort and HER2 expression levels (*BRCA1* promoter, hypermethylation, *PIK3CA* mutations, *PTEN* deletions, CXCR2 levels) [14,15,16,17].

### 2.3. Survival Analyses

Using 19 November 2020 as the cut-off date, the median follow-up was 9.7 years (95% CI [9.2; 10.4]). During this period, 88 deaths and 72 relapses were recorded. Therefore, the 5-year overall survival (OS) was 80.5%, 95% CI [75.4; 84.6], and the 5-year relapse-free survival (RFS) was 78.0%, 95% CI [72.7; 82.4]. Most relapses occurred during the first years of follow-up, which is in agreement with the previously reported temporal distribution of relapse risk in patients with TNBC [1,18].

The univariate analysis used to identify variables associated with RFS and OS (Table 3) found that classical prognostic factors, such as age, tumor stage, programmed death-ligand 1 (PD-L1) expression by tumor cells, TIL presence, and molecular apocrine phenotype, were significantly associated with survival. Adjuvant chemotherapy was also associated with better RFS and OS. The impact of HER2 expression level on OS and RFS is summarized in Figure 1. HER2 2+ expression (N = 8) was associated with worse RFS compared with HER2 0/1+ (hazard ratio (HR): 3.16, 95% CI 1.27–7.85, *p* = 0.034; Figure 1F). However, no statistical association was found by stratifying patients according to other HER2 expression level groups (0 vs. 1+ vs. 2+; 0 vs. 1+/2+).

In the multivariate analysis (Table 4), tumor size, node invasion, histological type, molecular apocrine phenotype, TIL infiltration, and adjuvant chemotherapy were significant determinants of OS. Nodal status, TIL infiltration, and adjuvant chemotherapy were significantly associated with RFS.

## 3. Discussion

TNBC remains an unmet medical need due to the scarcity of validated targets. In the absence of recurrent druggable targets, sustained efforts have been made to refine its taxonomy, in order to discover therapeutic opportunities [19,20,21]. Recently, it has been suggested that low HER2 expression level could be a potential target [22], particularly after the results of the DESTINY-Breast01 study [13]. Moreover, guidelines have been issued to better define the HER2-low subgroup of breast tumors [6]. However, the biology of breast cancers expressing low HER2 levels remains poorly investigated, especially in the TNBC group.

In this study, we evaluated a large and homogeneous cohort of patients after surgery for localized TNBC, without neoadjuvant treatment, to detect differences in clinicopathological variables and survival in function of HER2 expression levels. This cohort of patients has been extensively characterized in previous publications [14,15,16,17], allowing a comprehensive evaluation of the impact of HER2 expression levels in TNBC patients. The main finding is that HER2 1+/2+ TNBC were associated with the molecular apocrine phenotype, a specific TNBC subgroup with worse prognosis [15].

The biology of HER2-low TNBC remains poorly described. In the pooled analysis of individual patients (n = 2310 patients with breast cancer from four German neoadjuvant trials), 47.5% of the tumors were HER2 1+/2+ [11]. This percentage varied in function of the hormone receptor expression status as follows: 61.2% of hormone receptor-positive tumors and 34% of TNBC were HER2 1+/2+. Nearly the same percentage (36.6% of HER2 1+/2+ TNBC) was found by Schettini et al., who evaluated publicly available data on 3689 HER2-negative breast tumors [23].

These percentages are higher than the 16.2% of HE2 1+/2+ tumors observed in the present study. However, in our series, the HER2 scoring was reviewed following the 2018 ASCO/CAP guidelines, to obtain a homogeneous evaluation of the entire cohort. Conversely, in the previous pooled analysis, HER2 status was determined at the moment of inclusion in the clinical trial, with the inclusion periods extending from July 2012 to March 2019 (data not shown for the publication by Schettini et al.). This might explain part of the differences. Indeed, while the initial ASCO/CAP guidelines recommended a 30% cut-off for HER2 3+ tumors [24], the revised 2013 ASCO/CAP guidelines defined a threshold of 10% of strongly stained cells for HER3 3+ tumors [25]. Finally, as several laboratories reported a significant increase in the number of equivocal cases, the recommendations were updated in 2018, focusing mainly on the HER2 2+ definition [4]. As our cohort included samples collected between 2002 and 2012, we wanted to obtain comparable information on the HER2 status. Therefore, all IHC analyses were performed with the same antibody and followed the current recommendations for HER2 status interpretation.

Recently, Agostinetto et al. evaluated the transcriptomic profile of the HER2-low group using the PAM50 intrinsic subtype classification and data on 804 primary breast cancers (n = 410 HER2 1+/2+ tumors, among which 74 were TNBC) from the TCGA dataset [12]. They found that the percentage of HER2-enriched tumors (i.e., hormone receptor-negative and HER2-positive) was higher in the HER2 1+/2+ TNBC group (13.7% versus 1.6% in the HER2 0 TNBC population), unlike the study by Schettini et al. (9% of HER2-enriched tumors in the HER2 0 TNBC population vs. 7.1% in the HER2 1+/2+ TNBC population) [23]. A separate assessment of the HER2 1+ and 2+ populations could be used to determine the possible existence of a progressive gradient of HER2 pathway activation in this population.

While an association between the molecular apocrine subtype and HER2 overexpression (3+) has been previously reported [26], our results suggest that molecular apocrine tumors are more frequently HER2-low than HER2 0 cancers. This association, and the recent description of FOXA1 as a determinant of chemotherapy resistance in breast cancer cells [27], strengthen the rationale of trials to evaluate the combination of androgen receptor-targeting agents with new-generation anti-HER2 targeted therapies.

Moreover, in our population, the HER2 2+ TNBC subgroup was small (2.7% of all TNBC analyzed), but presented worse RFS. This result expands the corpus of evidence on the prognostic impact of a low expression level of HER2, although the current literature shows conflicting results. In 2012, Rossi et al. reported, in a prospective study on 1150 patients operated on for breast cancer (88% hormone receptor-positive tumors), worse RFS for tumors with HER2 2+/FISH-negative status (n = 116 patients, among whom 29 had TNBC) compared with HER2 0/1+ tumors. Therefore, the authors suggested that these tumors could be candidates for anti-HER2 treatment [28]. The phase III NSABP-B47 study, testing the combination of trastuzumab with adjuvant chemotherapy in localized cancers, did not find any benefit of trastuzumab addition in the HER2-low population [8]. However, it detected a significant interaction between HER2 expression levels and treatment group (*p* = 0.03). In the HER2 2+ subgroup (stratification factor), trastuzumab addition was associated with an HR of 2.28 (95% CI 1.21–4.31) for OS. This difference disappeared after adjustment for histological grade and treatment assignment. In a retrospective study on 5097 patients with localized breast cancer, HER2 2+ tumor expression was also a poor prognostic factor for RFS (*p* < 0.0001) compared with HER2 0/1+ in the hormone receptor-positive (PR and/or ER) subgroup, but not in the TNBC subgroup (n = 654) [10]. In the study by Agostinetto et al., no difference was found between the HER2 0 and HER2 1+/2+ groups, in terms of disease-free interval, progression-free interval, and OS, after stratification in function of the hormone receptor status [12]. Similarly, no difference in OS was found based on HER2 status in the study by Schettini et al. [23]. In the neoadjuvant setting, HER2 1+/2+ breast tumors displayed lower pathological complete response rates compared with HER2 0 tumors in the German pooled analysis (29.2% vs. 39.0%, *p* = 0.0002) [11]. However, this difference was not observed in the TNBC subgroup (50.1% vs. 48.0%, *p* = 0.21). No specific evaluation was reported for the HER2 2+ subgroup. Altogether, the published evidence suggests a detrimental impact of HER2 2+ expression level in early TNBC, and advocates the need for dedicated studies on the rare HER2 2+ subtype, in terms of biology, prognosis, and therapeutic targeting.

Our findings, if validated in an independent series, could open the way to the evaluation of combined anti-HER2 and androgen receptor-targeted strategies in this population with unmet medical needs.

## 4. Materials and Methods

### 4.1. Objectives

The primary endpoint of the study was the evaluation of the impact of HER2 expression level (0, 1+, 2+) on survival (OS and RFS).

Secondary endpoints were the evaluation of the association between HER2 expression level and baseline clinicopathological variables.

### 4.2. Patients and Tumor Samples

Our tumor biobank (Biobank number: BB-0033-00059) includes samples and data of 1695 consecutive patients with histologically proven breast cancer who were treated at our center between 2002 and 2010. These frozen tumor samples were mainly used for ER and PR assessment with the dextran-coated charcoal (DCC) method, as previously described [29,30], or for uPA/PAI-1 quantification (Femtelle^®^ test). Breast cancer samples with ER/PR concentration <10 fmol/mg of protein (DCC assay), or with <10% of ER/PR-positive tumor cells (IHC), were classified as ER- and PR-negative [31]. HER2 protein expression was evaluated by IHC with the A485 polyclonal antibody (Dako, Glostrup, Denmark). Tumors with HER2 scores of 0 and 1+ were classified as HER2-negative. In samples with HER2 score of 2+, gene amplification was assessed by ISH using the HER2 IQISH pharmDx kit (Dako, Glostrup, Denmark) for HER2 copy number and chromosome 17 centromere detection. Tumor samples with HER2 3+ scores were classified as HER2-positive. Tumor HER2 status was reviewed following the 2018 ASCO/CAP guidelines for HER2 scoring [4].

For this analysis, data on 296 patients with primary non-metastatic TNBC, without neoadjuvant treatment, were used. Each patient was managed in accordance with our institution guidelines [32]. Their clinicopathological characteristics and treatment(s) are summarized in Table 1. This study was approved by the Montpellier Cancer Institute Institutional Review Board (ID number: ICM-CORT-2020-28). All patients signed a written, informed consent form. As part of the study evaluated the prognostic impact of biological markers, this manuscript adheres to the REMARK guidelines.

### 4.3. Tissue Microarray and Immunohistochemistry

Tissue block selection from the Biological Resource Center of the Montpellier Cancer Institute (Biobank number: BB-0033-00059), tissue microarray construction, IHC analyses and scoring (EGFR, CK5/6, PD-L1, AR, FOXA1) have been previously published [14,15,16].

### 4.4. Statistical Analysis

Percentages were used to describe categorical variables, and medians and ranges were used for continuous variables. Categorical variables were compared with the Pearson’s chi-square or Fisher’s exact test. OS and RFS were defined as the time from the surgery date to death (any cause) and to tumor recurrence, respectively. The date of the last recorded visit and the date of death were used to censor patients who were alive at the last follow-up without recurrence/loss to follow-up and patients who had died without recurrence, respectively. OS and RFS rates were calculated with the Kaplan–Meier method and compared with the log-rank test. The Cox proportional hazard model was used for multivariate analysis of all variables with a *p*-value < 0.2 in univariate analysis. HR are presented with the 95% CI. STATA 13.0 (StatCorp, College Station, TX, USA) was used for all statistical analyses.

## 5. Conclusions

In a large, homogeneous series of patients with early TNBC, HER2 1+/2+ expression is associated with a lower histological grade and a molecular apocrine profile, and a specific subgroup of androgen receptor-expressing TNBC with worse long-term prognosis. Considering the current development of anti-androgens for MA tumors and anti-HER2 ADC for this HER2 1+/2+ population, this clinicopathological association could have therapeutic implications. Although rare (2.7% of patients in our series), HER2 2+ TNBC seems to have worse RFS, advocating a dedicated clinical, biological and therapeutic evaluation of this subgroup.

## Figures and Tables

**Figure 1 cancers-13-06059-f001:**
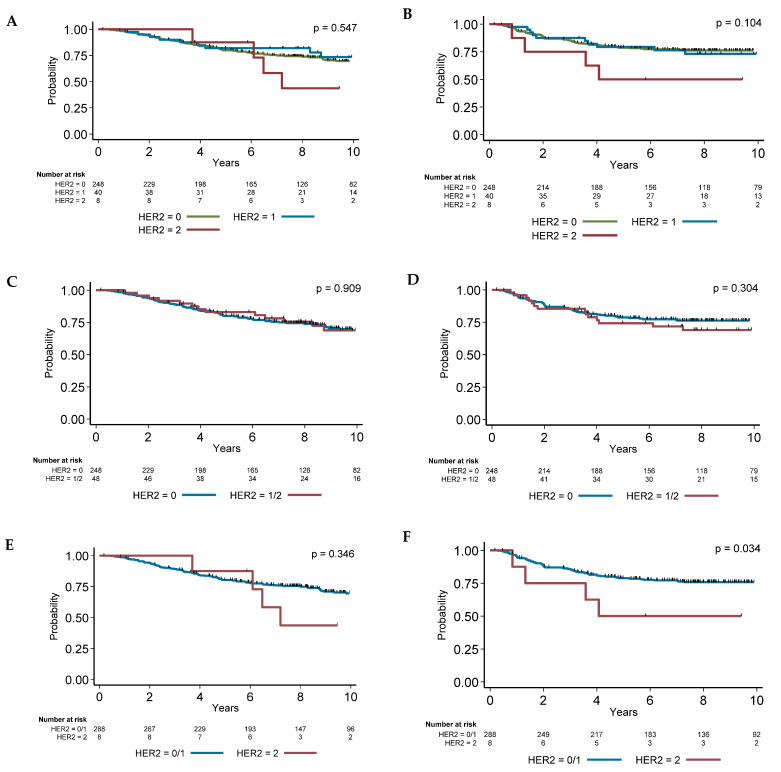
Overall survival (**A**,**C**,**E**) and relapse-free survival (**B**,**D**,**F**) in function of HER2 expression level.

**Table 1 cancers-13-06059-t001:** Patient and tumor characteristics.

Variables	N = 296	%
Age (years), median [min–max]	57.7 [28.5–98.6]	
Tumor size (missing = 1)		
T1	135	45.8
T2	139	47.1
T3/T4	21	7.1
Nodal status		
N−	188	63.5
N+	108	36.5
Histological grade (missing = 6)		
1–2	61	21
3	229	79
Histology (missing = 3)		
Ductal	240	81.9
Lobular	16	5.5
Other	37	12.6
Adjuvant chemotherapy (missing = 1)		
No	73	24.7
Yes	222	75.3
Basal-like phenotype (missing = 3)		
No	106	36.2
Yes	187	63.8
Molecular apocrine (AR/FOXA1) phenotype (missing = 20)		
AR+/FOXA1+	111	40.2
AR−and/or FOXA1−	165	59.8
TILs %, median [min-max] (missing = 10)	**5**	**(1–80)**
≤5	174	60.8
>5	112	39.2
PD-L1 expression tumor cells (%; missing = 26)		
<1%	116	43
≥1%	154	57
PD-L1 expression immune cells (%; missing = 29)		
0	50	18.7
]0–10]	80	30
]10–50]	75	28.1
>50	62	23.2
HER2		
0	248	83.8
1+	40	13.5
2+	8	2.7

AR: androgen receptor, FOXA1: forkhead box protein A1, N−: Node negative, N+: Node positive, TILs: tumor-infiltrating lymphocytes. Basal-like phenotype was defined by positive staining for cytokeratin 5/6 and/or EGFR (≥1% tumor cells stained in IHC), while its absence was defined by the lack of cytokeratin 5/6 and EGFR staining.

**Table 2 cancers-13-06059-t002:** Univariate correlations between TNBC features and HER2 expression level by IHC.

Variables	HER2 = 0		HER2 = 1+/2+		*p*-Value
	N	%	N	%	
Age (years), median [min–max]					**0.047**
<55	116	46.8	15	31.3	
≥55	132	53.2	33	68.7	
Tumor size					
T1	115	46.6	20	41.7	
T2	114	46.1	25	52.1	0.803
T3/T4	18	7.3	3	6.2	
Node status					
N−	157	63.3	31	64.6	0.866
N+	91	36.7	17	35.4	
Histological grade					**0.007**
1–2	44	18.2	17	35.4	
3	198	81.8	31	64.6	
Basal-like phenotype					0.387
No	86	35.1	20	41.7	
Yes	159	64.9	28	58.3	
Molecular apocrine (AR/FOXA1) phenotype					**0.008**
AR+/FOXA1+	84	36.7	27	57.4	
AR−and/or FOXA1−	145	63.3	20	42.6	
TILs%					0.948
≤5	145	60.9	29	60.4	
>5	93	39.1	19	39.6	
PD-L1 expression in tumor cells (%)					0.236
<1	94	41.4	22	51.2	
≥1	133	58.6	21	48.8	
PD-L1 expression in immune cells (%)					0.382
0	38	17	12	28	
]0–10]	69	30.8	11	25.6	
]10–50]	65	29.	10	23.2	
>50	52	23.2	10	23.2	

AR: androgen receptor, FOXA1: forkhead box protein A1, N−: Node negative, N+: Node positive, PD-L1: programmed death ligand-1, TILs: tumor-infiltrating lymphocytes. Basal-like phenotype was defined by positive staining for cytokeratin 5 and 6 and/or EGFR (≥1% tumor cells stained by IHC).

**Table 3 cancers-13-06059-t003:** Univariate analysis of clinical pathological variables in function of OS and RFS.

Variables	OS			RFS		
	HR	95% CI	*p*-Value	HR	95% CI	*p*-Value
Age (years)			**<0.001**			0.090
<55	1			1		
≥55	2.18	1.38–3.46		1.51	0.93–2.44	
Tumor size			**<0.001**			**<0.001**
T1	1			1		
T2/T3/T4	2.92	1.80–4.74		2.52	1.51–4.23	
Nodal status			**<0.001**			**<0.001**
N−	1			1		
N+	2.07	1.36–3.15		3.87	2.38–6.29	
Histological grade			0.442			0.714
1–2	1			1		
3	0.82	0.51–1.34		0.90	0.52–1.55	
Histology			0.063			0.582
Ductal	1			1		
Other	0.57	0.30–1.08		0.84	0.45–1.57	
Adjuvant chemotherapy			**<0.001**			**0.003**
No	1			1		
Yes	0.29	0.19–0.44		0.46	0.29–0.75	
Basal-like phenotype			0.646			0.443
No	1			1		
Yes	1.11	0.71–1.73		0.83	0.52–1.33	
Molecular apocrine phenotype			**0.043**			**0.035**
AR−and/or FOXA1−	1			1		
AR+/FOXA1+	1.56	1.02–2.40		1.66	1.04–2.65	
TILs%			**0.002**			**<0.001**
≤5	1			1		
>5	0.47	0.29–0.78		0.40	0.23–0.70	
PD-L1 expression tumor cells (%)			**0.041**			**0.029**
<1%	1			1		
≥1%	0.63	0.41–0.98		0.59	0.36–0.95	
PD-L1 expression immune cells (%)			0.171			**0.025**
0	1			1		
]0–10]	1.57	0.82–3.01		1.41	0.73–2.73	
]10–50]	0.87	0.42–1.80		0.49	0.21–1.12	
>50	0.94	0.45–1.98		0.85	0.40–1.81	
HER2 (0 vs. 1+ vs. 2+)			0.547			0.104
0	1			1		
1+	0.83	0.43–1.61		1.06	0.54–2.07	
2+	1.64	0.60–4.50		3.19	1.28–7.95	
HER2 (0 vs. 1+/2+)			0.909			0.304
0	1			1		
1+/2+	0.97	0.55–1.71		1.36	0.77–2.40	
HER2 (0/1+ vs. 2+)			0.346			**0.034**
0/1+	1			1		
2+	1.68	0.62–4.59		3.16	1.27–7.85	

AR: androgen receptor, FOXA1: forkhead box protein A1, N−: Node negative, N+: Node positive, TILs: tumor-infiltrating lymphocytes. Basal-like phenotype was defined by positive staining for cytokeratin 5/6 and/or EGFR (≥1% tumor cells stained in IHC), while its absence was defined by the lack of cytokeratin 5/6 and EGFR staining.

**Table 4 cancers-13-06059-t004:** Multivariate analysis to identify variables associated with OS and RFS.

Variables	OS (N = 263)			RFS (N = 285)		
	HR	95% CI	*p*-Value	HR	95% CI	*p*-Value
Tumor size			**<0.001**			
T1	1					
T2/T3/T4	2.56	1.49–4.38				
Nodal status			**0.003**			**<0.001**
N-	1			1		
N+	2.02	1.26–3.23		4.43	2.68–7.30	
Histology						
Ductal	1		**0.002**			
Other	0.35	0.16–0.73				
Adjuvant chemotherapy			**<0.001**			**0.001**
No	1			1		
Yes	0.34	0.21–0.53		0.40	0.24–0.65	
Molecular apocrine phenotype						
AR-and/or FOXA1-	1		**0.048**			
AR+/FOXA1+	1.58	1.00–2.49				
TILs%			**0.027**			**0.002**
≤5	1			1		
>5	0.57	0.35–0.96		0.44	0.25–0.77	

AR: androgen receptor, FOXA1: forkhead box protein A1, TILs: tumor-infiltrating lymphocytes.

## Data Availability

The data presented in this study are available on reasonable request from the corresponding author.

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
