# Peer review of "Prognostic Value of HER2-Low Expression in Non-Metastatic Triple-Negative Breast Cancer and Correlation with Other Biomarkers"

_cancers, 2021, doi:10.3390/cancers13236059_

Round 1

Reviewer 1 Report

In this work the author evaluate the impact of HER2 espression level and its association with clinicopathological features and survival outcomes in a cohort of patients with early TNBC treated and characterized for emerging biomarkers. The published evidences suggest a detrimental impact of HER2 2+ expression level in early TNBC and advocate the need of dedicated studies on the rare HER2 2+ subtype in term of biology, prognosis and therapeutic targeting.
In my opinion the work is well written and no revisions are required. 

Author Response

Thank you fo your positive evaluation and your interest in this work.

Following your comment on the English language and style item, a double check has been performed by a native English speaker.

With my best regards

Reviewer 2 Report

I enjoyed reading your work. It is neat and precise research updates in Her2+ TNBC. Only suggestion would be seeking validation of this interesting results in multi-center studies in the future. As the authors pointed out, "Our findings, if validated in an independent series, could open the way to the evaluation of combined anti-HER2 and androgen receptor-targeted strategies in this population with unmet medical needs." This would be a breakthrough in treatment regimen and quite clinic-relevant. If validated, more studies and clinical trials would be warranted in this rare but important subpopulation of HER2+ TNBC patients.

The writing of this research report is straightforward and easy to follow for the readers. In addition, the clinical descriptions are accurate, and the statistic models are correct. Very good.

Author Response

(The authors gave the same response as above.)

Reviewer 3 Report

In my opinion this work is very interesting and clear, the results are convincing with clinical-therapeutic applicability. Even from a statistical point of view it seems to me to be well conducted. it is a good idea to evaluate the weight of HER2 expression in triple negative tumors, in particular with reference to immunohistochemical expression of androgen receptors. I also think that demonstrating that even a low expression of HER 2 (HER2 low) has a prognostic and therapeutic impact, will push pathologists to an increasingly accurate evaluation of HER2 negative cases (distinguish 0 from 1+).

Author Response

Thank you fo your positive evaluation and your interest in this work.

With my best regards

Reviewer 4 Report

The manuscript examined the prognostic value of HER2-low expression in non-metastatic TNBC retrospectively in 296 patients. They found that the HER2 1+/2+ tumors exhibited lower histological grade and MA profile and found more frequently in older patients. They also found that HER2 2+ tumors were associated with worse Relapse-Free Survival in comparison with HER2 0/1+ tumors, although Overall Survival (OS) and RFS were not different in the HER2 0 and 1+/2+ groups. 

The manuscript is well written and the interpretation of the data is reasonable, logical and convincing. The results are important for future treatment strategies for HER2-low TNBC.

Author Response

Thank you fo your positive evaluation and your interest in this work.

Following your comment on the English language and style item, a double check has been performed by a native English speaker.

With my best regards

This manuscript is a resubmission of an earlier submission. The following is a list of the peer review reports and author responses from that submission.